# Improved Gridded Precipitation Data Derived from Microwave Link Attenuation

Micha Silver [1,*,†,‡] , Arnon Karnieli [1,†,‡] and Erick Fredj [2,‡]

1   Jacob Blaustein Institutes for Desert Research, Ben-Gurion University of the Negev, Sede-Boker 84990, Israel; karnieli@bgu.ac.il
2   Computer Science Department, The Jerusalem College of Technology, Jerusalem 91160, Israel; fredj@jct.ac.il
*   Correspondence: silverm@post.bgu.ac.il; Tel.: +972-523-665-918
†   Current address: Remote Sensing Laboratory, Jacob Blaustein Institutes for Desert Research, Ben Gurion University, Beer Sheva 84105, Israel.
‡   These authors contributed equally to this work.

**Abstract:** The motivation for improving gridded precipitation data lies in weather now-casting and flood forecasting. Therefore, over the past decade, Commercial Microwave Link (CML) attenuation data have been used to determine rain rates between microwave antennas, and to produce more accurate countrywide precipitation grids. CML networks offer a unique advantage for precipitation measurements due to their high density. However, these data experience uncertainty from several sources as reported in earlier research. This current work determines the reliability of rainfall measurements for each link by comparing CML-derived rain rates to adjusted weather radar rainfall at the link location, over three months. Dynamic Time Warping (DTW) is applied to the pair of CML/radar time-series data in two study areas, Israel and Netherlands. Based on the DTW amplitude and temporal distance, unreliable links are identified and flagged, and interpolated gridded precipitation data are derived in each country after filtering out those unreliable links. Correlations between CML-derived grids and rain observations from an independent set of gauges, tested over several rain events in both study areas, are higher for the reliable subset of CML than the full set. For certain storm events, the Kendall rank correlation for the set of reliable CML is almost double that of the complete set, demonstrating that improved gridded precipitation data can be obtained by removing unreliable links.

**Keywords:** commercial microwave links; weather radar; dynamic time warping; interpolation

## 1. Introduction

Underlying the efforts to improve gridded precipitation data are the needs of weather now-casting and flood forecasting. Flash flooding can lead to unfortunate consequences with extensive economic damages and loss of life (Gaume et al. [1], Merz et al. [2], Ward et al. [3], Pappenberger et al. [4]). Accurate hydrological modeling can assist in avoiding these negative effects, which, even a few hours of early warning, in the framework of now-casting (Franch et al. [5], Heuvelink et al. [6]), are enough to mobilize emergency staff and to evacuate populations when necessary. However, this level of forecast accuracy is attained only when the model inputs are accurate, where first and foremost are detailed and precise precipitation data. Chwala et al. [7] showed an operational, real-time system for producing Commercial Microwave Link (CML) based precipitation data for hydro-meteorological applications in Germany. A similar effort in the Netherlands was described by Overeem et al. [8] and in Israel by David et al. [9]. In a conference presentation, Gosset et al. [10] discussed the use of CML data for rainfall measurements in gauge-poor regions of Africa.

The unique advantage of CML networks for producing gridded precipitation data are rooted in the high density of antennas and links (Bianchi et al. [11]). Commercial

cellphone service providers typically install backhaul antennas spaced at a few kilometers apart throughout the service region. Research by Overeem et al. [8] and Zinevich et al. [12] reported average link densities of 7 and 8 links/100 km², respectively. On the other hand, rain gauge density is typically an order of magnitude lower (Kidd et al. [13]). Essentially, rain rate data derived from a CML network can enhance or replace an existing set of rain gauge observations, as demonstrated by Gosset et al. [10]. Using accepted geo-statistical methods such as kriging (Goovaerts [14]) or inverse distance weighted (Kurtzman et al. [15]), the CML-derived rainfall can be interpolated to create spatially explicit gridded precipitation data. Furthermore, weather radar grids can be successfully adjusted with CML-derived rain data as shown in Sideris et al. [16], Foehn et al. [17], and Silver et al. [18] by interpolation using kriging with external drift.

Signal strength between CML antennas is affected by rainfall causing attenuation of the broadcast power (Messer [19], Goldshtein et al. [20]) at the receiving antenna. Attenuation of the signal results from absorption and scattering by airborne water droplets, similar to the effect utilized by weather radar to identify rainfall. The relationship between signal attenuation and rain rate is described by a power law (detailed by Messer [19], Leijnse et al. [21]), as:

$$A = \alpha \cdot R^{\beta} \qquad (1)$$

where $R$ represents the rain rate (mm/h) and $A$ is the resulting path averaged attenuation (dB/km). The two parameters $\alpha$ and $\beta$ are empirical and depend on both microwave frequency and the distance between the microwave antennas. The International Telecommunications Union publishes standard tables for $\alpha$ and $\beta$ values over ranges of frequency and path length (as recommendation P.838-3 in https://www.itu.int/rec/R-REC-P.838-3-200503-I/en, accessed on 15 June 2021). The A-R relation is very nearly linear at frequencies near 35 GHz, where the $\beta$ exponent approaches 1, while, below 35 GHz, the $\beta$ value drops.

However, CML signal attenuation and the resulting rain rate are affected by several sources of uncertainty such as the wet radome effect, determining a baseline attenuation to distinguish between wet and dry periods, and variations in the Drop Size Distribution (DSD) along the length of the link. These uncertainties, explained in Leijnse et al. [22], Berne and Uijlenhoet [23], Leijnse et al. [24], are not necessarily random. Instead, they may reflect systematic differences among the links. For example, the distance between antennas, known as path length, can influence attenuation due to non-meteorological interference in the atmosphere, such as dust or fog that increase with distance. Scattering of microwaves that impact hydro-meteors changes with DSD that, in turn, might also vary along the path length. Both Zinevich et al. [25] and David et al. [9] mentioned variation in DSD along the link as a source of uncertainty. In a carefully controlled experiment, Fencl et al. [26] reported that uncertainty in CML-derived precipitation grew during periods of heavy rain. Thus, links between distant antennas, crossing uneven topography, or located in areas susceptible to dust, fog, or localized rain patterns could suffer from irregular changes in attenuation, unrelated to meteorology, resulting in rainfall estimation errors. Recent work by Kim and Kwon [27] indicated that weather radar could be successfully adjusted by using data from CML links specifically in the nearby vicinity of each radar pixel. Furthermore, Rios Gaona et al. [28] investigated uncertainty in CML-derived rainfall by comparing both real and simulated link rainfall with weather radar. They suggested that removing specific, problematic links could improve the resulting precipitation grid.

Building on past research that identified and developed the methodology to retrieve rain rates from CML attenuation, this work strives to improve interpolated rain grids by correctly identifying problematic links and filtering out those data. The one-time process of identifying those unreliable links relies on a time-series analysis of CML-derived rainfall compared to weather radar rain grids, using Dynamic Time Warping (DTW). The enhanced rain grids are validated against rainfall observations from independent sets of gauges.

## 2. Methods and Materials

### *2.1. Study Area and Data Sets*

This research was conducted in two diverse study areas: central and northern Israel, and the Netherlands (Figure 1). The Israeli study area experiences a Mediterranean climate, and annual average rainfall of 200 mm in the arid south to a maximum of 900 mm in the mountainous north, concentrated in the winter months. The Netherlands typically receives around 900 mm annual precipitation, more or less evenly distributed throughout the year. The topography in northern Israel is mountainous, with peaks of over 1000 m, whereas the Netherlands is uniformly low in elevation.

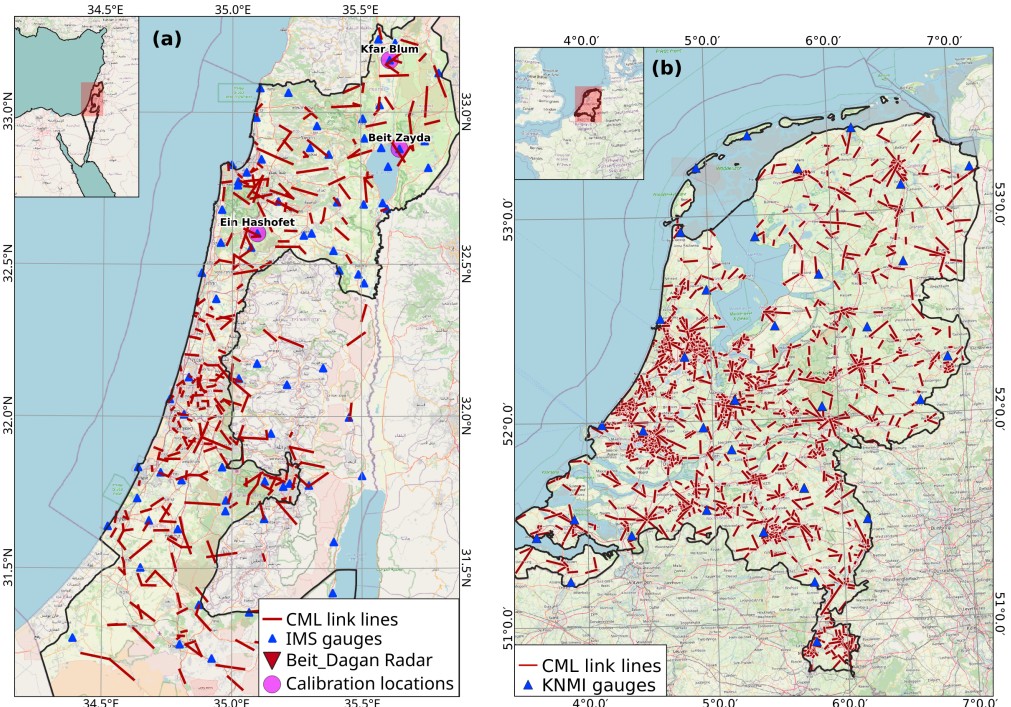

**Figure 1.** Two study areas (**a**,**b**) with CML links from cellphone providers and gauge locations. The Israel study area includes calibration locations (Section 3.1.1).

### 2.1.1. CML Data

Among the cellular phone service providers in Israel, Pelephone Ltd. offered CML attenuation data for 324 links across the country (red lines in Figure 1) and covering three winter (rainy) seasons, from fall 2016 to spring 2019. Received signal power data were available at 15-min time intervals. Approximately 45% of the links operate at 23.3 Ghz, and about 40% at around 18 Ghz, with the remainder using frequencies between those. The length of the links varied from less than 1 to over 15 km, with the majority of links in the range below 6 km. Figure 2a shows both the distribution of path lengths and (b) various microwave frequencies used over the range of path lengths.

CML attenuation data for the Netherlands were acquired from the 4TU.ResearchData https://data.4tu.nl/info, (accessed on 15 June 2021) data center. This public data-set covers two periods: from 9 June 2011 to 11 September 2011 (with some gaps) and from 30 May 2012 to 12 September 2012 (continuous). Attenuation data from over 2700 microwave links are available at 15 min time intervals, and with minimum/maximum power for each interval. The number of links and span of path lengths are both larger than the Israel network, and many links operate at the 38 MHz frequency (Figure 2c,d). A majority of links are bi-directional, thus the data-set contains attenuation in both directions between many pairs of links.

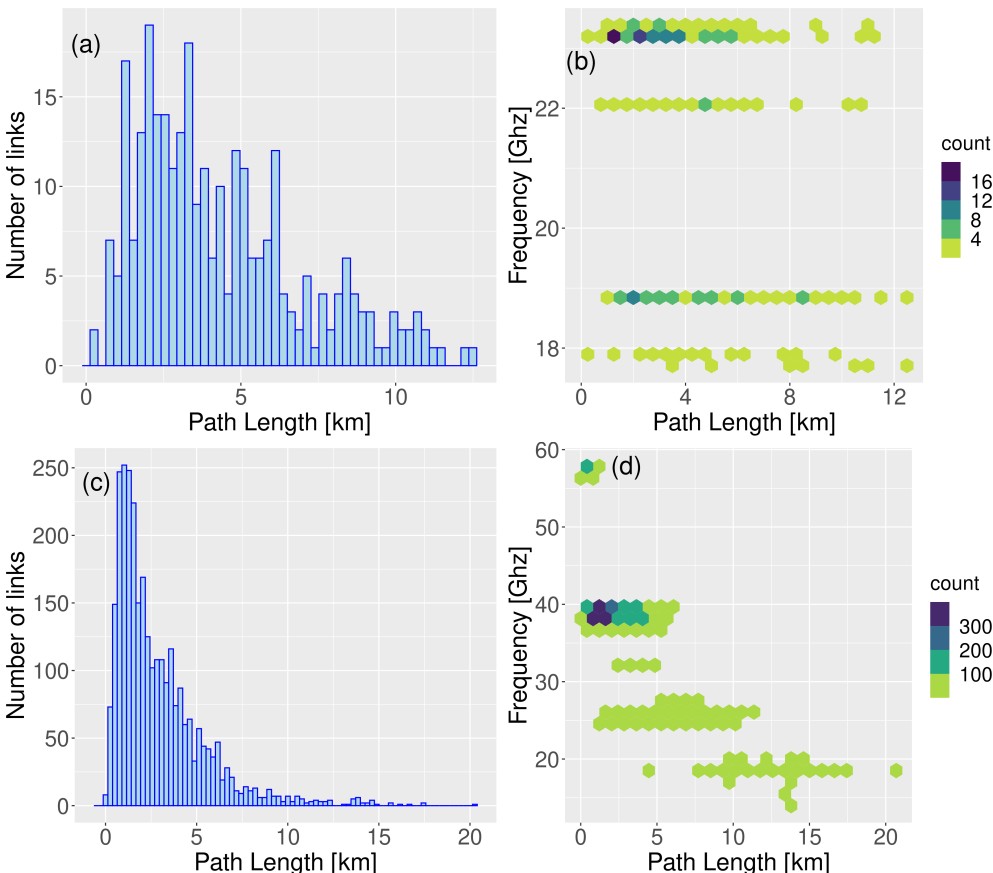

**Figure 2.** Distribution of path lengths and frequencies for CML networks in Israel (**a**,**b**), and the Netherlands (**c**,**d**). The histogram bin width is 250 m, so each column represents the number of links with lengths within a 250 m span. The Netherlands network consists of a greater number of links, higher frequencies, and some longer path lengths.

### 2.1.2. Radar Data

The Israeli Meteorological Service (IMS) operates a C-band, single polar weather radar at the Beit Dagan installation (the red triangle on Figure 1). Processed images from this radar, projected onto a 1 km Cartesian grid, are publicly available (on request from IMS) in two formats: raw uncorrected images and adjusted images using the Integrated Now-casting through Comprehensive Analysis (INCA) protocol (Kann et al. [29], and presented in http://www.zamg.ac.at/fix/INCA_system.pdf, accessed on 15 June 2021). This protocol applies a multiplicative Inverse Distance Weighted (IDW) adjustment to the raw radar using the IMS network of rain gauges. The raw and adjusted products are available at three temporal aggregations: 10 min, hourly, and 24 h. For this work, the INCA adjusted, 24-h images were chosen, since the IMS considers these images as best representing true rainfall (based on personal communication). Their confidence in this product is based on two different reasons: 24-h aggregations tend to overcome uncertainties in fast moving storms (discussed in Marra and Morin [30]); this 24-h aggregation employs the full set of 400 gauges for adjustment (see below), whereas the hourly aggregations use only a subset of about 80 automatic stations.

A public archive of corrected radar images for the Netherlands is available from the Royal Netherlands Meteorological Institute (KNMI), catalogued at https://dataplatform. knmi.nl/dataset/, accessed on 15 June 2021. In this work, precipitation accumulations for hourly, climatological gauge-adjusted radar images at 1 km resolution, data-set name `rad_nl25_rac_mfbs_01h`, were acquired and used.

### 2.1.3. Gauge Data

IMS maintains a countrywide network of approximately 380 meteorological stations of which about 20% (the blue triangles in Figure 1) of these stations communicate data continuously at 10-min intervals, and are used by IMS for real-time weather forecasting. The IMS then aggregates these data into hourly, daily, and annual rainfall. Data from the remaining stations are transferred daily at about 6:00 a.m. UTC. Daily gauge aggregations from the automatic stations were obtained and used for the validation stage in this current work.

Archived gauge data for the Netherlands at hourly time intervals are available also from the KNMI data catalog. Data from 39 automatic stations overlapping the period of the Netherlands CML data-set (Section 2.1.1) were acquired.

### 2.2. Deriving Rainfall from CML Attenuation

The R package RAINLINK (Overeem et al. [31] was used to transform CML attenuation data to rain rates. This package implements a tool chain to convert received power levels to rain rate, including procedures to determine the reference "dry" base-line level of attenuation, removing of wet antenna effects, extracting the $\alpha$ and $\beta$ parameters from standardized tables and application of the A-R power law relationship (Equation (1)). Power levels at the receiving antenna are typically recorded by cellular service providers in one of two ways, either minimum and maximum during a short time interval, or received power at a "snapshot" in time. The RAINLINK package successfully handles both types. The data from Israel in this work contained received power levels only at snapshots every 15 min. On the other hand, the Netherlands CML data consisted of minimum and maximum recorded power, also during 15-min intervals.

Rain rate was calculated for each 15 min data time step, where path length and frequency for each individual link were used to extract the A-R relationship parameters, $\alpha$ and $\beta$. RAINLINK includes a procedure to distinguish between wet and dry periods for application of the wet antenna attenuation parameter. For the Netherlands study area, the threshold parameters required by this wet-dry procedure were adopted from de Vos et al. [32].

However, de Vos et al. [32] suggest that wet antenna attenuation should be adjusted for different climates, and present a procedure to determine the optimal value for this wet antenna $Aa$ parameter. In the Israel study area of this work, their procedure was implemented as follows: three calibration locations (see Figure 1) were identified where CML links were close to IMS rain gauge stations (less than one kilometer perpendicular distance from the link line, and less than two kilometers from the link center point). CML rainfall rates were derived at hourly aggregations over four storm periods of three to five days each for these three links. Rain rate derivation was repeated with $Aa$ values varying from 0.4 dB to 4.0 dB. These rainfall rates were then compared to gauge observed, true rainfall rates from the nearby station to find the optimal $Aa$ value.

### 2.2.1. Dynamic Time Warping

Dynamic Time Warping (DTW), one of the algorithms for measuring similarity between temporal sequences, is an accepted approach in pattern recognition or signal processing, and has been applied to a meteorological analysis by Gilleland and Roux [33] and Mantas et al. [34]. DTW compares both amplitude and time lag between two time-series of data. The algorithm allows for matching events that are close in either time, amplitude, or both. DTW distance values were calculated by applying the R package dtw (Giorgino [35]) to the two time-series: CML rain rates and gauge observations. Since the number of data points in both data sets was equal, the Sakoe–Chiba window type was chosen, and, following the early work by Sakoe and Chiba [36], a small window size of 3 was chosen. The step pattern was left at the default "symmetric2".

### 2.2.2. CML-Derived Rainfall, Countrywide

With the chosen `RAINLINK` parameters (see Section 3.1.1), 15-min rain rates for all links in both study areas were calculated and then aggregated. For the Israel case study, total CML rainfall for 24-h time intervals covering the three-year study period was obtained. Since all IMS radar images and gauge data of daily aggregations begin and end at 6:00 a.m. UTC, the 24-h aggregations of CML data were also prepared from 6:00 a.m. to 6:00 a.m. UTC the following day, to match the IMS timing. The `RAINLINK` outlier removal procedure was not activated in this work; instead, outlier identification methods were applied to flag unreliable links for derivation of rainfall. Outlier removal within `RAINLINK` filters individual data points at a single time slot. In contrast, the approach adopted here searches along a time-series of data compared to an independent source of rain data, as explained below in Section 2.5.

For the Netherlands case study, hourly aggregations were prepared covering short rain events. Unlike the Israel case, where IMS radar images were available only at daily aggregations, the KNMI data center includes hourly, historic radar images, as well as gauge observation data. Thus, the analysis in the Netherlands was conducted at the higher temporal resolution, allowing for taking full advantage of the DTW approach.

### 2.3. Correlation to Radar Images

Since `RAINLINK` outputs path averaged rainfall along the link lines, comparison with weather radar rainfall required averaging several radar pixels (at 1 km resolution in both study areas) that overlapped the link line. While Rios Gaona et al. [28] extracted proportional rainfall from radar pixels partly covered by a link, a different approach was chosen in this current work. Referring to the Israel case study, 24-h aggregations of both radar images and CML-derived rainfall were used. Thus, the effects of fast-moving storm fronts that partially cover a radar pixel at short time intervals were assumed to become irrelevant. However, it should be mentioned that Fencl et al. [26] showed increased uncertainty over longer aggregation periods when matching CML and rain gauge measurements. Nevertheless, referring to the enlarged radar image in Figure 3, the extraction of rainfall from the radar images required averaging several radar pixels for each link. In this way, rainfall from signal attenuation between two microwave towers was compared to the average rainfall from all radar pixels intersected by the link line.

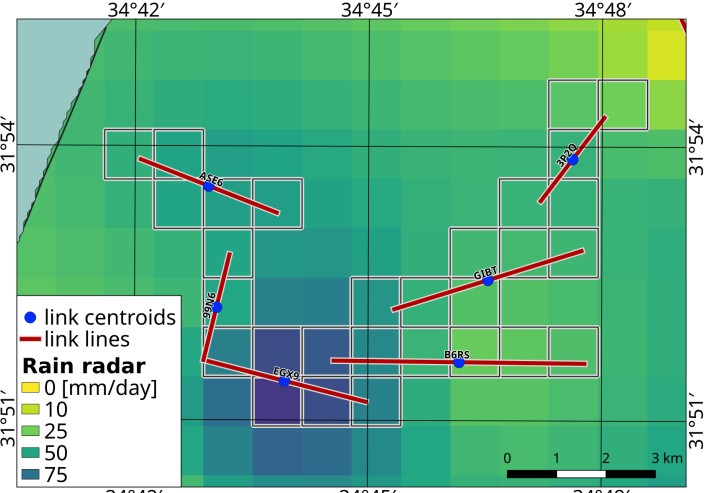

**Figure 3.** Zoomed image of weather radar over the northern part of Israel. Radar image values are summed for all pixels intersected by each link line. This sum of radar rainfall is assumed to be located at the link centroid for kriging interpolation.

Comparisons between CML-derived rainfall and the IMS weather radar for four sample links appear in Figure 4. Plots (a) and (b) are links that display consistent errors, where the link in (a) underestimates, and (b) overestimates the radar rainfall. Plots (c) and (d), on the other hand, show links that display irregular errors compared to the radar. During the period displayed in Figure 4, there were several significant storm events. Some links in the study area recorded rain on up to 18 separate days, while some links had fewer rain days. These links with fewer than four rain days were removed from the analysis, since such a small number of data points was too few for any correlation analysis, leaving 256 links with sufficient data.

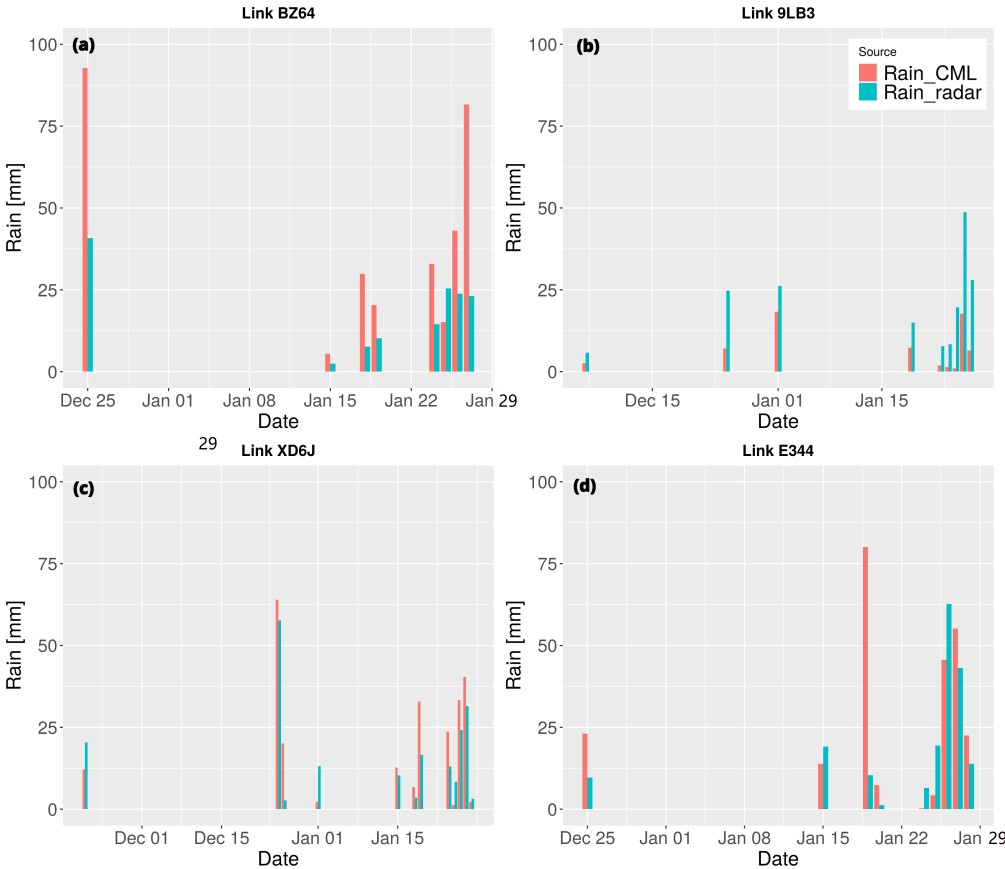

**Figure 4.** Four example links from Israel, with time-series comparisons between CML-derived rainfall and radar rainfall at the link location. The comparison period is from 1 November 2018 to 28 February 2019. Columns represent rainfall in mm/day. In (**a**), the link consistently over estimates daily rainfall; (**b**) shows a link that consistently underestimates rainfall. Plots (**c**,**d**) are links that display inconsistent errors compared to the radar images.

### 2.4. CML-Radar Correlation, Countrywide

With radar image values, extracted from all pixels along the length of each link, two time-series of rain measurements were available for each link—both the CML-derived rainfall and the radar image values "under" those links representing 24-h radar aggregations. In the Israel case study, a 24-h aggregation period was chosen for both data-sets, as discussed in Section 2.1, to overcome high variability in fast moving storms. DTW was applied to these two time-series of daily aggregations despite this long aggregation time, using the `dtw` function parameters as in Section 2.2.

These time-series were compared over the three-month calibration period from November 2017 to January 2018. The result of applying this algorithm to two data sets, the DTW distance value, will be referred to hereafter as DTWd. Thus, after applying DTW

to the time-series pair, each individual link gained a DTWd value that indicated how closely the CML-derived rainfall matched the radar image rainfall during the three-month period.

The data availability from KNMI in the Netherlands allowed for retaining a high temporal resolution of hourly CML-derived and radar precipitation. Thus, the DTW approach was particularly suited to this case study. Three months of CML attenuation data at 15 min intervals were aggregated to hourly derived rain rates. Radar precipitation images from June to September of 2012 served as the calibration data, also at hourly intervals. The DTWd parameter was calculated for all links using this extensive, high-resolution data.

### 2.5. Removal of Unreliable Links

The next stage identified those links that resulted in unreliable rainfall values. In statistical terms, this translated to outlier detection among the DTWd values. Extremely large DTWd values occurred when the rain amplitude or timing between CML and radar, over the calibration period, diverged. Therefore, those links with extreme DTWd values were considered unreliable for rain measurement.

In their paper, Rios Gaona et al. [28] also compared radar rainfall to CML-derived rainfall. They performed CML-radar correlations at high temporal resolution (15 min) and also compared simulated CML rainfall to a window of radar pixels surrounding the links. Their work suggested that removing erroneous link data, whether due to instrument failure or long path lengths, improved correlations. In contrast, this current work examines actual CML-derived rainfall to perform CML-radar comparisons over a long time-series, and filtering of unreliable links relied on DTW (Berndt and Clifford [37]).

An initial examination of the distributions of DTWd values, shown in Figure 5, indicated that they were not Gaussian, but rather matched fitted lognormal or gamma curves for Israel and Netherlands, respectively. These distributions are characterized by high probability density for low variable values, and a long tail of high variable values with low density. The fit between the actual DTWd values and a lognormal or gamma distribution indicates that most of the links attain a good match (low DTWd values) between the CML-derived rainfall and the known radar rainfall over the calibration interval. In contrast, a few links diverge and give unreliable rainfall measurements.

However, considering these non-Gaussian distributions of DTWd, a straightforward z-score filtering, based on two standard deviations from the mean, would not have been applicable. Instead, outliers were identified using two non-parametric outlier detection methods: Interquartile Range (IQR) and Median Absolute Deviation (MAD). These methods do not rely on a normally distributed population or the distribution mean.

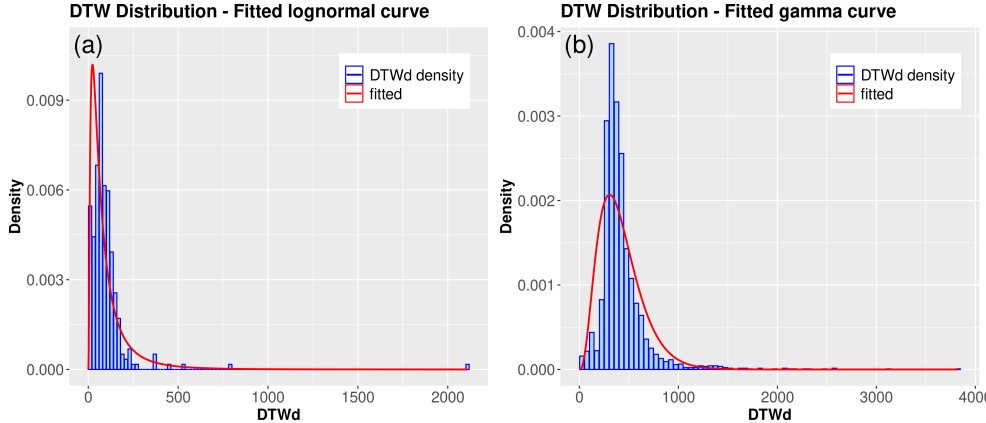

**Figure 5.** Probability density functions of DTWd values. The red lines are fitted distributions: (**a**) the DTW distance distribution for the CML network in Israel shows a good match to a lognormal distribution. In (**b**), the distribution of DTW distance values in Netherlands is shown, with a fair match to a gamma distribution. In both case studies, the best fitted distributions are exponential in nature, not Gaussian, thus justifying the choice of the IQR outlier filtering method.

The IQR, defined as $IQR = Q_3 - Q_1$, is the difference between the third quartile and the first quartile. Typically, data points are flagged as outliers when their value is greater than $Q_3 + 1.5 \cdot IQR$. In this work, a more aggressive filtering was chosen: DTWd values that exceeded $Q_3 + 1.0 \cdot IQR$ were flagged as unreliable. The additional outlier detection test, $MAD$, is calculated as in Equation (2):

$$MAD = median(abs(x_i - median(X))) \cdot 1.4826 \tag{2}$$

for a vector $X$ of values $x_i$. The scaling constant 1.4826 is set to $1/Q_3$ or the inverse of the third quartile in a standard normal distribution. This scaling constant ensures that the expected value of $MAD$ is approximately equal to the standard deviation. Thus, the $MAD$ value is scaled such that outlier detection is similar to using a regular "two standard deviations" for normal distributions. Then, outliers are those values $x_i$ where (Equation (3)):

$$\frac{abs(x_i - median(X))}{mad(X)} > 2 \tag{3}$$

Interestingly, both outlier detection methods flagged more or less the same set of links.

### 2.6. Validation

Once the DTWd values were calculated and outliers flagged, spatially explicit gridded precipitation data were prepared for both study areas. Kriging interpolation was applied to each of two sets of links: the **Full** set, and the filtered, **Reliable** set (with outlier flagged links removed). The CML-derived data represented path averaged rainfall between the microwave towers, along the length of the link. Nevertheless, the interpolation procedures assumed that the rainfall was located at the center-point between the CML towers following Overeem et al. [8] and Overeem et al. [31]. Zinevich et al. [12] presented rainfall interpolations derived from a tomographic analysis of multiple crossing links, and Goldshtein et al. [20] distributed the CML-derived rainfall at point locations along each link, also accounting for crossing links. Both of their methods relied on a high density of links, and therefore are primarily applicable in urban areas. A GIS procedure was performed on the **Full** data set of link lines in Israel to identify crossing link lines. Only 64 crossing links were found among the total of 324 (about 20%). Thus, the tomographic approach was not relevant in this study area, and the simple link center-point method was then chosen for both study areas.

### 2.6.1. Validation, Israel

Gridded precipitation data were created for the Israel study area by interpolating the CML-derived rainfall using three interpolation methods: Ordinary Kriging (OK), Kriging with External Drift (KED), and Inverse Distance Weighted (IDW) all implemented in the `automap` (Hiemstra et al. [38]) package in R. While the OK interpolation was based only on the CML rainfall, the KED algorithm merged the CML rainfall with a second precipitation grid which served to determine the trend. Raw, uncorrected radar images were obtained from IMS, at 24-h aggregations, to be used as the trend. As explained in Section 2.1.2, corrected radar images were also available, but these were used to calculate DTWd (Section 2.4) and then to distinguish unreliable links. Therefore, in order to avoid bias in the validation, KED was performed using the uncorrected radar images. IDW is typically more robust, since it does not depend on an estimated variogram model (Chen et al. [39]). However, predicted precipitation at locations distant from all CML tends to be less accurate (Haberlandt [40]). The resolution of the interpolated grid in all cases was set to 1 km to match the radar images.

The three interpolation algorithms were applied to each of eight storm events that occurred between October 2016 and March 2019, for 24-h intervals covering the storm, thus creating eight sets of 24-h precipitation grids. Then, in parallel, daily aggregated rain gauge observations were obtained from IMS covering those same 24-h periods. Rainfall

from the CML interpolated grids was extracted at all IMS gauge station locations, using a single pixel from the interpolated grid. No averaging of pixels surrounding the gauge station locations was applied. Rather, rainfall from each station was compared to one pixel from the interpolated precipitation grid. Then, correlation between these two sets of rain data was tested using Kendall's $\tau$. This rank-based, non-parametric correlation test does not depend on a normal distribution of the tested variable residuals. The choice of this algorithm stemmed from the non-Gaussian distribution of rainfall residuals among the gauge locations.

Rain gauge data from a total of almost 400 gauge stations were available; however, each storm event had measurable rainfall for only a subset of the gauges (refer to column 8 in Table 1). Thus, the 24 December 2017 storm, for example, recorded rainfall at 381 stations, and the event of 17 February 2018 measured rainfall at 394 gauges. The 27 October 2016 event, on the other hand, used only 118 stations.

**Table 1.** Correlation (Kendall's $\tau$) in Israel. Correlations between CML-derived rainfall and gauge observations, using Ordinary Kriging, Kriging with External Drift and Inverse Distance Weighted are presented. Both the **Full** set of CML and **Reliable** (outliers removed) set are presented for each interpolation method, as well as the number of gauges used for each event.

| Method: | OK | | KED | | IDW | | Number |
| Date | Full | Reliable | Full | Reliable | Full | Reliable | Gauges |
| --- | --- | --- | --- | --- | --- | --- | --- |
| 27 October 2016 | 0.143 | 0.112 | 0.258 | 0.114 | 0.304 | 0.369 | 118 |
| 24 December 2017 | 0.183 | 0.374 | 0.207 | 0.375 | 0.279 | 0.299 | 381 |
| 27 January 2018 | 0.113 | 0.323 | 0.129 | 0.341 | 0.395 | 0.386 | 263 |
| 17 Febuary 2018 | 0.195 | 0.219 | 0.199 | 0.233 | 0.160 | 0.154 | 394 |
| 13 November 2018 | 0.377 | 0.372 | 0.377 | 0.372 | 0.454 | 0.462 | 396 |
| 7 December 2018 | 0.192 | 0.213 | 0.410 | 0.390 | 0.276 | 0.266 | 303 |
| 16 March 2019 | 0.239 | 0.243 | 0.222 | 0.223 | 0.263 | 0.233 | 374 |
| 30 March 2019 | 0.261 | 0.295 | 0.254 | 0.328 | 0.290 | 0.283 | 394 |

2.6.2. Validation, Netherlands

In the Netherlands study area, OK was used to produce interpolated, countrywide rain grids. The higher link density in this study area suggested that ordinary kriging would be optimal. The target resolution was set to 1 km again to match the available gauge-adjusted radar images from the KNMI catalog. Two interpolation runs were performed: one using the **Full** CML network, and the second using only the **Reliable** CML links (after removal of the outliers). These interpolations were applied to hourly CML aggregations over the three-month validation period, June, August, and September of 2011.

Next, gauge data also at hourly temporal resolution were obtained from the KNMI data center. Data for 39 automatic stations were available for the validation period. CML-derived rainfall at all gauge locations was extracted from both interpolations, **Full** and **Reliable**, hour by hour throughout the validation period. Thus, the Netherlands study area contained a much larger set of data points for correlation analysis than the Israel study area. Two sets of the Kendall $\tau$ tests were prepared: the **Full** CML, as well as the **Reliable** set of CML, compared to gauges.

To test the robustness of this method, and the CML network overall, a further examination was done in the Netherlands study area. Three percentages of CML data were selected randomly and removed from the **Full** set, 10%, 30%, and 60% of the total number of links. Then, these reduced CML data-sets were also interpolated as above, and rainfall values extracted at the 39 gauge locations, to test correlation with the gauge observations.

**3. Results**

The figures and tables below present comparisons in both study areas between gauge observations and CML-derived rainfall at the gauge locations for several rain events. Initially, individual CML links are presented, comparing CML-derived rainfall to nearby

gauge observations (Israel study area), and CML rainfall to radar images (Netherlands study area). Then, countrywide analyses are presented for both study areas.

### 3.1. Individual CML Locations

3.1.1. CML-Gauge Correlations at Three Calibration Locations, Israel

In a preliminary step, to determine the optimal wet antenna attenuation parameter for `RAINLINK` (Section 2.2), correlations between observations from three specific IMS gauges and CML-derived rainfall from a nearby link were evaluated. Results from these three calibration locations (Figure 1) appear in Table 2, including DTWd values along with the Kendall's $\tau$ correlation for a range of $Aa$ values.

**Table 2.** Correlation (Kendall's $\tau$) and DTW distance between gauge observations and CML-derived rain rates at hourly aggregations for three links. The evaluation is repeated for seven different wet antenna attenuation parameters $Aa$. These three stations are close to link lines, and this table compares gauge observed rain rates to CML-derived from the nearby link. Both Kendall's $\tau$ and DTW distance are averaged over four storm periods, to determine the optimal wet antenna parameter.

| Gauge | $Aa$ (dB) | Kendall's $\tau$ | DTW dist. |
|---|---|---|---|
| BEIT ZAYDA | 0.4 | 0.707 | 56.815 |
| BEIT ZAYDA | 0.8 | 0.740 | 49.388 |
| BEIT ZAYDA | 1.4 | 0.735 | 42.715 |
| BEIT ZAYDA | 2 | 0.649 | 45.481 |
| BEIT ZAYDA | 2.6 | 0.608 | 41.221 |
| BEIT ZAYDA | 3.2 | 0.561 | 42.176 |
| BEIT ZAYDA | 4 | 0.656 | 47.604 |
| EN HASHOFET | 0.4 | 0.651 | 70.762 |
| EN HASHOFET | 0.8 | 0.674 | 62.094 |
| EN HASHOFET | 1.4 | 0.682 | 58.919 |
| EN HASHOFET | 2 | 0.611 | 65.700 |
| EN HASHOFET | 2.6 | 0.593 | 65.142 |
| EN HASHOFET | 3.2 | 0.566 | 68.694 |
| EN HASHOFET | 4 | 0.610 | 67.652 |
| KFAR BLUM | 0.4 | 0.615 | 43.001 |
| KFAR BLUM | 0.8 | 0.646 | 31.484 |
| KFAR BLUM | 1.4 | 0.654 | 30.133 |
| KFAR BLUM | 2 | 0.593 | 35.218 |
| KFAR BLUM | 2.6 | 0.600 | 34.238 |
| KFAR BLUM | 3.2 | 0.535 | 35.734 |
| KFAR BLUM | 4 | 0.585 | 37.469 |

Using hourly aggregations of CML-derived and gauge data from the three chosen link/gauge pairs in the Israel study area, both DTW distance and Kendall rank correlation coefficient (Kendall's $\tau$) were calculated for each of the $Aa$ values in the range from 0.4 dB to 4.0 dB. This optimization step determined the wet antenna attenuation value that achieved the best match, i.e., highest Kendall's $\tau$ correlation and lowest DTW distance, over all four storm periods. Given the hourly rainfall aggregations, DTW was particularly suited to this comparison, since time warping can overcome any possible time lag between the locations of the CML antennas and the gauges. While the pairs of gauges and CML antennas were in close proximity, the dynamics of storm movement could confound a straightforward correlation even over short distances. DTW, on the other hand, accounts for time shifts due to storm movement.

As evident from in Table 2, maximum Kendall's $\tau$, together with minimum DTWd, appear at *Aa* value 1.4 dB for two of the three gauges evaluated. The BEIT ZAYDA gauge station revealed two minima of DTWd: at *Aa* 1.4 and also 2.6. The Kendall's $\tau$ correlation maximum for this gauge was at *Aa* 0.8, nearly in line with the other two gauge stations. Noting that this gauge station is located near the shore of the Sea of Galillee (Figure 6c), there might be a local micro climate, i.e., mists, that influences the CML attenuation, causing this anomaly. Thus, this iterative procedure covering four storm events indicated that wet antenna attenuation of 1.4 dB was optimal even in the relatively dry climate of the Israel study area. Figure 6 shows zoomed-in maps of the three CML links with nearby gauge stations, and plots of the rainfall over one storm event in January 2018, derived using the 1.4 dB antenna attenuation parameter.

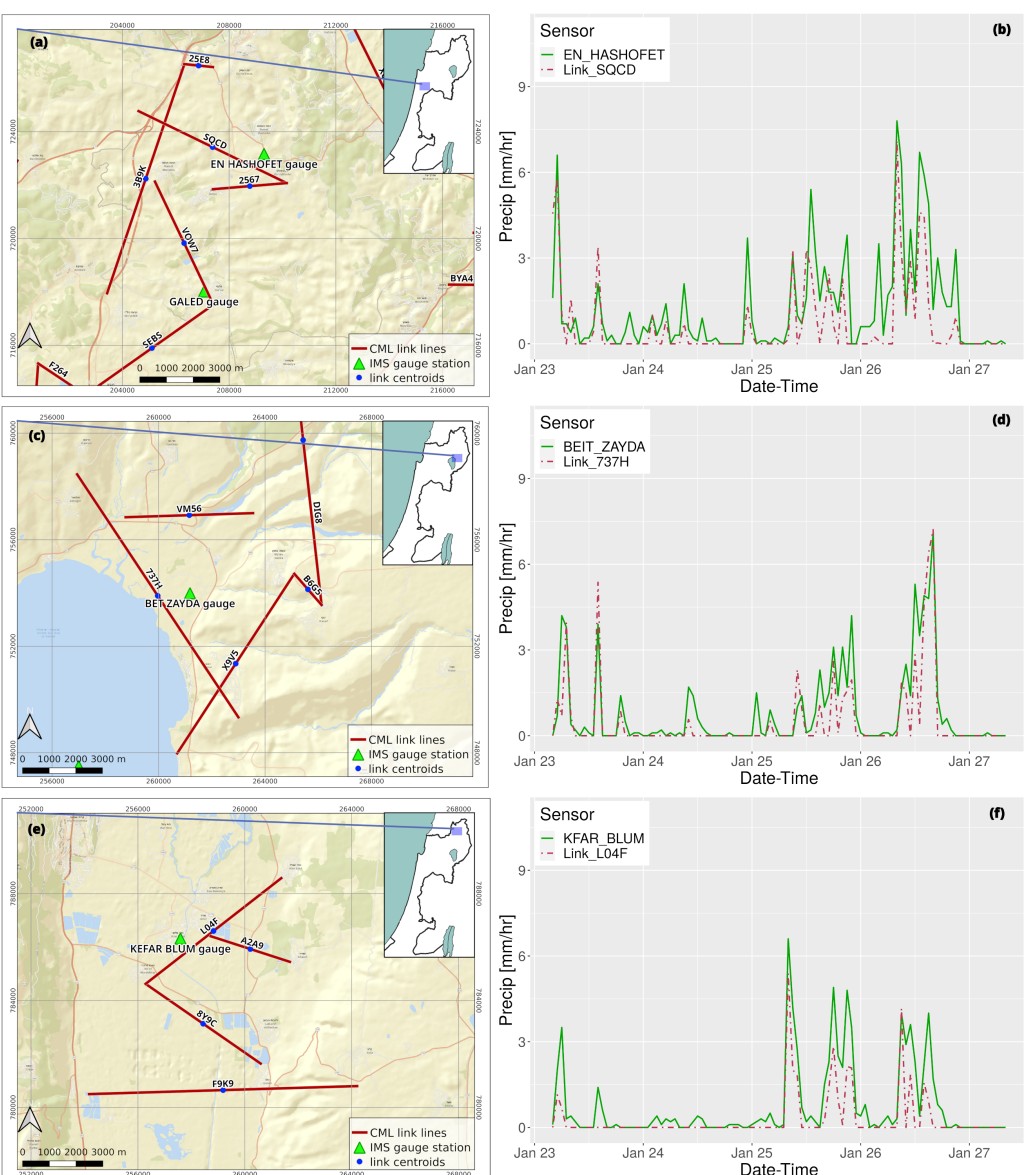

**Figure 6.** Comparison of hourly rainfall from three CML links and three nearby IMS gauges in northern Israel. Maps (**a**,**c**,**e**) show the locations of three particular IMS gauge stations, which are close to CML links. The graphs in (**b**,**d**,**f**) present hourly rainfall, where the solid green lines are the true gauge observations, and the red dashed lines represent CML-derived rain rates, both aggregated to millimeters per hour. These CML rain rates were derived using *Aa* of 1.4 dB.

### 3.1.2. CML-Gauge Correlations at Two Locations, Netherlands

An example comparison between CML-derived rainfall and weather radar precipitation for the Netherlands appears in Figure 7. CML rainfall for two links, over a period of three days in August, 2012, is shown in panels (a) and (b). These graphs also display the radar precipitation, extracted from pixels that intersect the link lines. The location of these particular links, and one radar image, are presented in panel (c) of Figure 7.

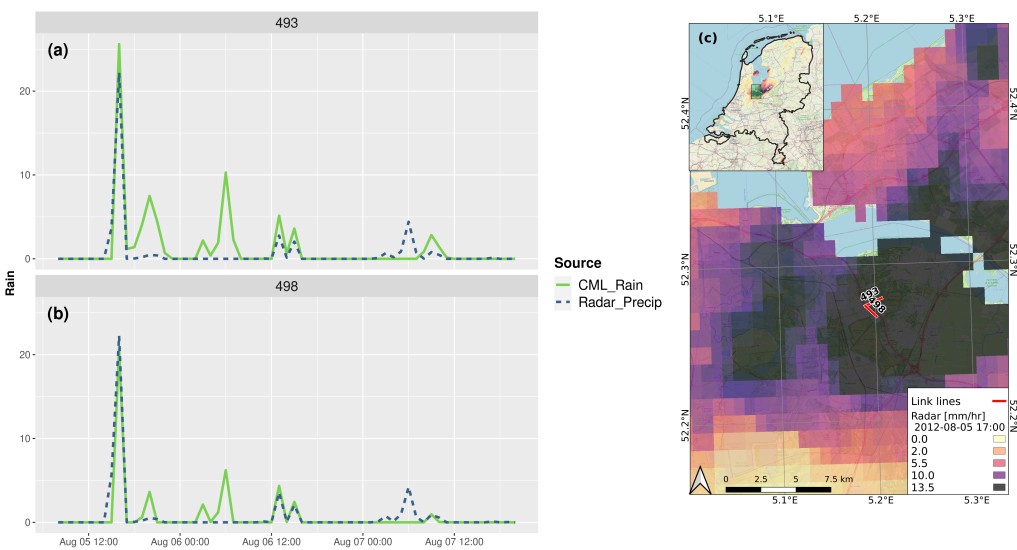

**Figure 7.** Comparison of hourly rainfall from two CML links and radar image in the central Netherlands. Graphs (**a**,**b**) show CML-derived hourly rainfall for three days with hourly radar precipitation at the two link locations. Map (**c**) displays the locations of these link lines, together with one hourly radar image, at 5:00 p.m. on 5 August 2012.

### 3.2. Countrywide Gauge-CML Correlations, Israel

The resulting precipitation grids for three storm events are shown in Figure 8 with gauge observations overlaid for visual comparison.

Table 1 presents Kendall's $\tau$ correlations between gauge observations and the CML-derived and interpolated precipitation grids, averaged over all gauges. On each of eight validation dates, a storm event passed over some part of the study area during the 24-h interval of the event. The first column contains the date of the storm event. For each of the three interpolation methods, the pair of columns shows correlations when the CML precipitation was interpolated using the **Full** set of available links and interpolation using only the **Reliable** links (those not flagged as outliers). The final column lists how many gauges recorded measurable rainfall during the 24-h interval for each storm event.

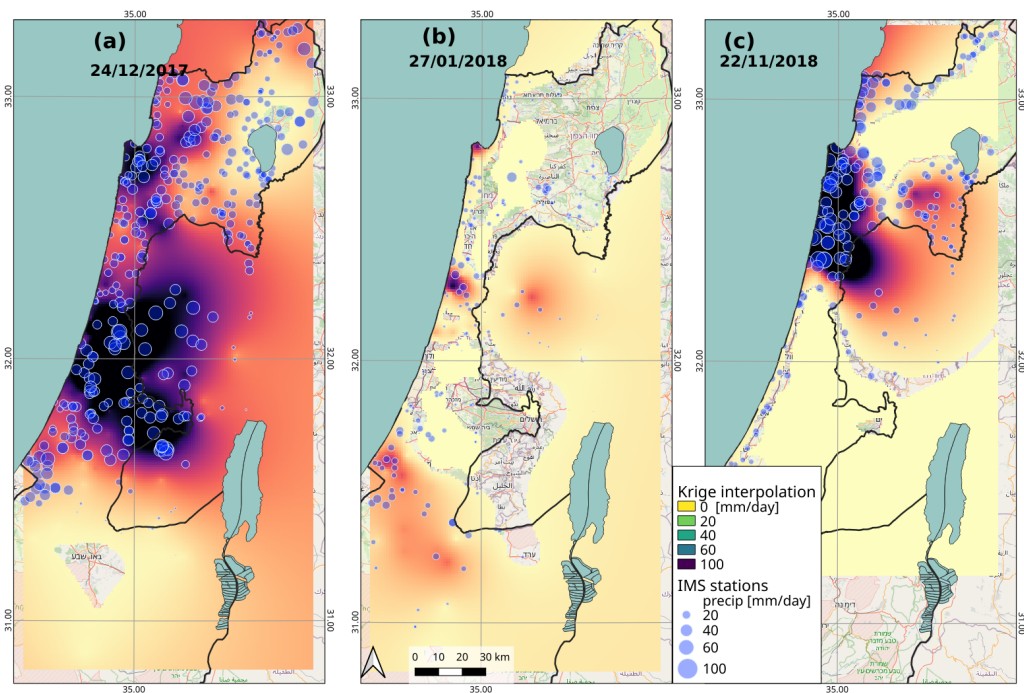

**Figure 8.** CML interpolated rainfall for three storm events. The interpolation method displayed is ordinary kriging. Gauge observed rainfall for the 24 h is overlaid as blue circles. Maps (**a–c**) show respectively the storm events for 24 December 2017, 27 January 2018, and 22 November 2018.

First, in seven among the eight storm events, the **Reliable** set of links (outliers removed) attained as good or better correlation to the gauge measurements as the **Full** set. In some cases, the difference in correlation was negligible (i.e., 23 November 2018), while, in some cases, such as 24 December 2017, 27 January 2018, and 30 March 2019 correlation from the **Reliable** CML was substantially higher. Next, surprisingly, OK achieved better correlation to the gauges than KED. Research papers on weather radar adjustment (Goudenhoofdt and Delobbe [41], Sideris et al. [16], Silver et al. [18]) often point to KED as the more successful adjustment algorithm. In this current work, the raw radar images, which were used as the trend for KED adjustment, were recognized by IMS as less accurate (personal communication) and, as such, caused shifts in the KED interpolation that led to somewhat lower correlation to the "true" gauge measurements compared to the simple OK interpolation. This radar data were chosen instead of the corrected radar images, as explained in Section 2.6, to avoid bias in the validation. Nevertheless, in some cases (30 March 2019, for example), the KED interpolation did attain better correlation to the gauge measurements.

The event of 27 October 2016 deserves special attention. The IMS characterized this storm as synoptic classification "Active Read Sea Trough" (ARST). (Their report, in Hebrew, appears online at: https://ims.gov.il/sites/default/files/%D7%A1%D7%A7%D7%99%D7%A8%D7%AA%20%D7%90%D7%99%D7%A8%D7%95%D7%A2%2028-27%20%D7%91%D7%90%D7%95%D7%A7%D7%98%D7%95%D7%91%D7%A8%202016.pdf, accessed on 15 June 2021). Storms of this synoptic class are often fast moving, highly convective, and unpredictable; they occur mostly in the fall or spring. In their paper describing the various synoptic classes in the Eastern Mediterranean, Alpert et al. [42] point out that only about 20% of the storm events develop from an ARST. In some cases, the movement of an ARST is such that the precipitation is partially or totally missed by weather radar or rain gauges. In this case, the CML-derived rainfall might actually represent more accurately the "true" rain, whereas the gauge-adjusted radar might have partly missed the storm, leading to the low correlation presented here. The authors do not have access to additional data that

would be required to verify this hypothesis. However, this storm event does spotlight the fact that filtering of a CML network for outliers should rely only on typical winter synoptic events. As described in Section 2.4, here, unreliable links were identified by analyzing typical Cyprus Low synoptic conditions in storm events from November 2017 through January 2018.

### 3.3. Countrywide Gauge-CML Correlations, The Netherlands

Figure 9 shows two sample interpolated grids: in panel (a), the interpolation used the **Full** set of CML, whereas, in panel (b), only the **Reliable** set (after removal of outliers) was used. Green circles indicate two small areas where the **Reliable** set corrected anomalies in the interpolation.

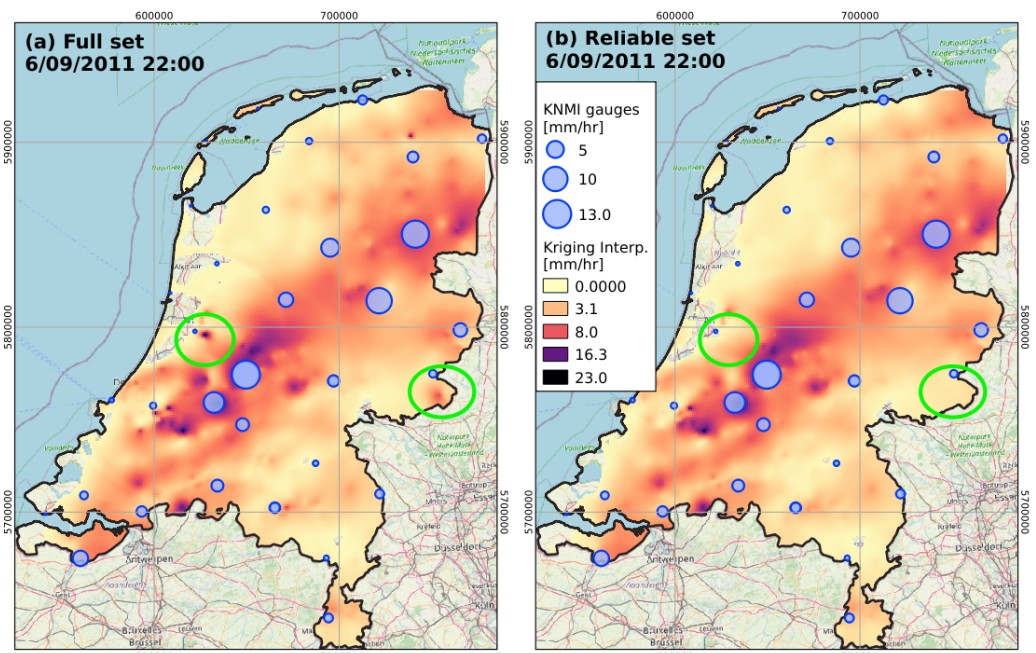

**Figure 9.** CML interpolated rainfall during a rain event in the Netherlands. The interpolation method displayed is ordinary kriging. Gauge observed rainfall for the one-hour period is overlaid as blue circles. Map (**a**) shows interpolation using the **Full** set of CML data, whereas in (**b**) only the **Reliable** CML are used. Two areas of corrections to the rain grid are encircled in green

To examine correlation between gauge observations and CML interpolated rainfall in the Netherlands, eight rain events during the three-month validation period were identified and isolated. These events (Table 3) extended from 7 to 30 h in duration, and maximum rain rates (countrywide) during the events were from 13 to 39 mm/h.

Correlations (Kendall's $\tau$) for five subsets of CML are shown in Table 4: the **Full** CML network, a subset with 10% randomly chosen links removed, another subset with 30% links removed, and the third subset with 60% removed, and finally the **Reliable** links (outliers removed). Among the eight rain events, the correlation for the **Reliable** set was higher than the **Full** set in three cases, and nearly identical to the **Full** set in three more cases. Only in two rain events was the **Reliable** correlation slightly lower than the **Full** set, and, in all events, the **Reliable** correlation was better than the subset of 10% random links removed. Two comparison scatter plots presented in Figure 10 display the correlation between gauges and CML interpolated rainfall at the gauge locations: the **Full** set and the **Reliable** subset. The data points represent 39 gauge locations, for each of the 22 h of the event duration.

**Table 3.** Rain events used for validation in the Netherlands.

| Event | | Duration | Rain Rate Maximum | Hours of Rain Median > 0.25 |
|---|---|---|---|---|
| From: | To: | [hours] | [mm/hour] | [hours] |
| 1 | 10 June 2011 12:00 | 10 Junuary 2011 21:00 | 10 | 30.5 | 0 |
| 2 | 6 August 2011 13:00 | 6 August 2011 22:00 | 10 | 16.4 | 3 |
| 3 | 8 August2011 00:00 | 9 August 2011 05:00 | 30 | 39.0 | 0 |
| 4 | 14 August 2011 01:00 | 14 August 2011 09:00 | 7 | 24.0 | 4 |
| 5 | 27 August 2011 09:00 | 27 August 2011 23:00 | 15 | 13.1 | 0 |
| 6 | 26 August 2011 07:00 | 26 August 2011 15:00 | 9 | 25.7 | 3 |
| 7 | 6 September 2011 10:00 | 7 September 2011 08:00 | 23 | 34.1 | 12 |
| 8 | 7 September 2011 12:00 | 8 September 2011 08:00 | 20 | 16.8 | 5 |

**Table 4.** Rain event correlations—Netherlands: **Full** set of CML, percent of randomly selected CML removed, and **Reliable** only (outliers removed).

| Event | Kendall $\tau$ | | | | |
|---|---|---|---|---|---|
| | Full Set | Random Removed 10% | 30% | 60% | Reliable |
| 1 | 0.604 | 0.591 | 0.587 | 0.540 | 0.623 |
| 2 | 0.588 | 0.581 | 0.571 | 0.539 | 0.586 |
| 3 | 0.585 | 0.569 | 0.573 | 0.514 | 0.580 |
| 4 | 0.702 | 0.690 | 0.683 | 0.665 | 0.699 |
| 5 | 0.555 | 0.549 | 0.543 | 0.478 | 0.550 |
| 6 | 0.523 | 0.513 | 0.530 | 0.520 | 0.517 |
| 7 | 0.731 | 0.732 | 0.732 | 0.722 | 0.742 |
| 8 | 0.603 | 0.599 | 0.594 | 0.555 | 0.612 |

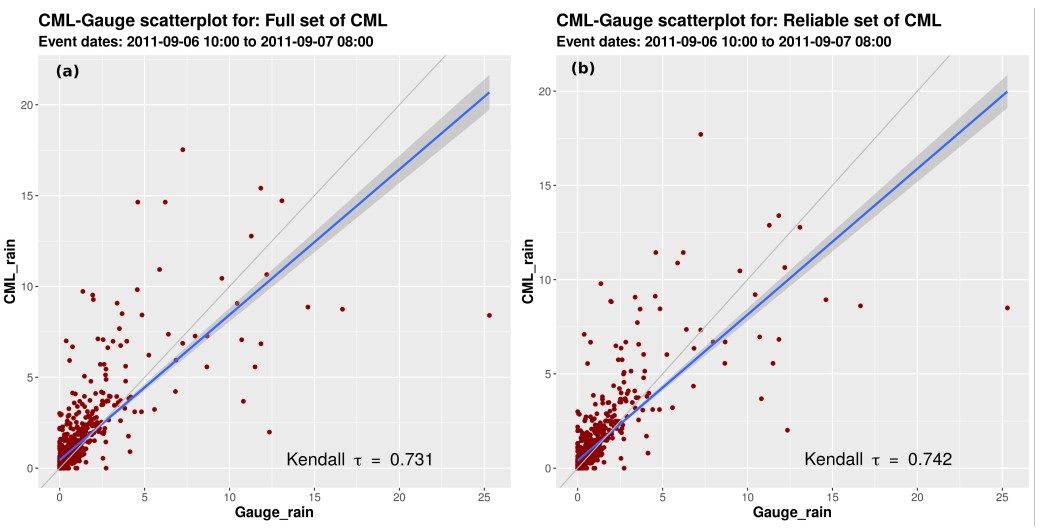

**Figure 10.** Comparison between 39 gauge observations and CML rainfall interpolation at the gauge locations over a 22-h duration rain event. Scatterplots of both the **Full** (**a**) and **Reliable** (**b**) sets of CML are shown.

## 4. Discussion

Building on the high density of CML networks, this work focuses on identifying microwave links that were unreliable for measuring rainfall, and demonstrates that precipitation grids can be improved by filtering those links from the CML network. Those links that systematically diverge from the radar were identified by implementing a time-series comparison between CML-derived rainfall and values from adjusted radar images at the link locations. Standard outlier detection procedures were applied to flag and filter out

those unreliable links. This one-time procedure resulted in a set of reliable links after removing outliers. Then, spatially explicit rainfall grids from the remaining links were prepared using geo-statistical interpolation to produce precipitation distributions for several storm events in both study areas. When the resulting precipitation grids were then correlated to independent sets gauge observations, a higher match was found over several storm events to precipitation derived from the filtered, **Reliable** set of CML data.

However, matching gridded precipitation to point observations is problematic, as described by both Overeem et al. [43] and Graf et al. [44]. This problem is compounded when deriving rainfall from CML attenuation, since the transformation of attenuation to rain rate results in path averaged rainfall, where the path length might be several kilometers. CML based precipitation grids in this work were obtained by assuming that the path averaged rainfall was located at the center-point between the pair of CML antennas (similar to the approach used by Overeem et al. [8]). Thus, link line center-points were used when interpolating the CML-derived rainfall to spatially explicit grids. This research addressed the issue of path averaged vs. point observations and gridded precipitation in two ways. First, regarding the Israel study area, when determining the optimal wet antenna parameter value, $Aa$, (Section 2.2), three specific gauges where chosen in close proximity to a link line. Then, gauge-observed and CML-derived rain rates were aggregated to hourly rainfall. While time lags could still occur as a storm passes from the link line to the nearby gauge, evaluation of the correlation between rain rates relied on DTW, which allowed for shifts between the two time-series. Then, in determining the outlier links (Section 2.4), both CML-derived rainfall and the reference radar grids were aggregated to 24-h periods, thus avoiding most time lag and storm movement issues. This long aggregation interval resulted in the high correlation between CML-derived rainfall and gauge observations (Table 1) since the spatial variance of rainfall decreases as the aggregation interval increases (Marra and Morin [30]). Furthermore, DTW was again applied to assess the match between path averaged rainfall for each link and the average rainfall of radar pixels touched by that link.

The second approach to overcome the problems with point observations relied on the high spatial and temporal density of data. In the Netherlands study area, the CML data were aggregated to hourly temporal resolution throughout. Furthermore, it is worth pointing out that the density of CML links in the Netherlands is much higher than the Pelephone data-set used in Israel. The density of links over the Netherlands (41,500 km$^2$), after merging bi-directional links to one data point and leaving 1850 link centroids, was approximately 4.4 links/100 km$^2$, whereas the smaller study area in Israel (19,000 km$^2$) was covered by only 256 links, giving a density of approximately 1.4 links/100 km$^2$.

The kriging interpolation method applied to data from both study areas relies on a variogram, the variance in data point values as a function of the distance between the points. Oliver and Webster [45] show that, as sample size of data points increases, the error in the experimental variogram decreases. Thus, with a large number of data points, the resulting kriging interpolation is less sensitive to shifted (or even missing) points. The Netherlands CML data should, therefore, produce more accurate rain grids. Examining the results in Table 4 for the CML sets with 10% or 30% randomly selected points removed reinforces that conclusion. Even with a third of the CML data removed, correlation with gauges degrades only slightly. Thus, the high density of links in the Netherlands leads to a robust network of rainfall measurements. Furthermore, the exact location of the CML-derived rainfall, whether at the link center-point, or distributed along the link line, should therefore not have influenced the resulting interpolated countrywide grids. Thus, the high number of links over the study area justified placing the path averaged rainfall at link centroids for kriging interpolation.

Another salient point is raised when considering the robustness of the Netherlands CML network to removal of random links. On one hand, the drop in correlation between gauge observations and the CML interpolation is minimal, even when a third of the points are removed. Then, similarly, the improvement in correlation when only **Reliable** (outliers

removed) links are used is also minor. Those small increases in Kendall's $\tau$ correlation for the **Reliable** set, in column 6 of Table 4, should therefore be considered significant, in light of the overall stability of the CML network.

## 5. Conclusions

Past research has found that rainfall causes attenuation of microwave signal strength, and that the degree of attenuation can be used to calculate the rain rate between microwave towers. With an eye towards now-casting, the high spatial and temporal density of microwave tower attenuation data in many countries can produce gridded precipitation data from CML networks. This work focused on identifying unreliable CML links, in order to improve the resulting precipitation data.

The relation between signal attenuation and rain rate is described by a power law (Equation (1)), where empirical parameters that depend on frequency and path length have already been determined. Commercial installations of microwave links supply cellular communication world-wide, and the high density of these CML networks has opened a new opportunity to produce interpolated precipitation grids in near real-time, for use in weather now-casting and flood forecasting. However, the CML-derived rain rates also involve a level of uncertainty. Causes for this uncertainty include variations in drop size distribution along the link paths, long path lengths, and non-meteorological atmospheric interference. Some of these causes might be systematic, leading to consistently high error levels for specific links.

This work demonstrated that those links that prove unreliable for measuring rain rates could be identified. Using dynamic time warping (DTW), a time-series comparison was created between CML-derived rainfall and adjusted weather radar as an independent and known source of precipitation data. The distribution of DTWd values among all links did not match a Gaussian distribution (Figure 5) so inter-quartile range and median absolute deviation methods were used to detect outliers. These accepted outlier detection methods were then applied using the DTWd outcome to flag and remove unreliable links.

With outlier links removed, the remaining, **Reliable** set of links was interpolated using geo-statistical procedures. Thus, spatially explicit and improved precipitation grids were obtained from the CML-derived rainfall. In order to validate that the remaining, **Reliable** links indeed improved precipitation grid results, and these precipitation grids were correlated to rainfall measurements from independent rain gauge data, acquired from national meteorological services. A comparison of gauge correlations (Tables 1 and 4) between the **Full** and **Reliable** sets of CML data indicated that the filtered, reliable-only interpolations more closely matched gauge measurements over most of the tested rain events in both study areas. Specifically in the Netherlands study area, the high density of CML and good temporal resolution of the data led to stable precipitation grids, resilient to missing data points. The correlation to gauge observations degraded only when more than half of the CML data were removed. This robust network, therefore, showed only slight improvements in interpolated precipitation grids when using the **Reliable** CML. However, that very resilience in the CML network should explain that those seemingly negligible improvements shown in Table 4 are, in fact, significant.

The question of the accuracy of CML-derived rainfall during atypical convective storms, such as ARST synoptic events in the Israel study area, remains open. Availability of additional data-sets of CML attenuation covering the southern arid region that is usually affected by ARST events could open a new line of research targeting these extreme events.

Improved precipitation grids can be obtained from CML attenuation data after carrying out the preparatory step of identifying outliers. A time-series comparison is required between CML-derived rainfall and some independent and known precipitation measurements over an extended calibration period. In the Israel study area, this calibration period was chosen to cover winter storms which develop from typical (Cyprus Low in this study area) synoptic conditions. In the Netherlands study area, rain events occur throughout the year, and the calibration period was selected based on availability of CML data sets.

Then, accepted outlier detection methods were applied to identify the subset of **Reliable** links, based on the time-series comparison. This subset of the CML network was then interpolated using geo-statistical methods to provide spatially explicit precipitation grids. Validation of these grids against an independent network of gauge observations established that improved precipitation grids can be produced from dense networks of CML-derived rainfall by removing unreliable links for rain measurement.

**Author Contributions:** Conceptualization, M.S., A.K. and E.F.; methodology, M.S. and E.F.; software, M.S.; validation, M.S.; formal analysis, M.S.; data curation, M.S.; writing—original draft preparation, M.S.; writing—review and editing, M.S., A.K. and E.F.; visualization, M.S.; supervision, A.K. and E.F.; project leader, E.F.; funding acquisition, E.F. All authors have read and agreed to the published version of the manuscript.

**Funding:** This research was funded by the Karlsruhe Institute of Technology IMV-IFU under the "Integrating Microwave Link Data For Analysis of Precipitation" (IMAP) grant (DFG Grant No.: GZ: KU2090/7-2, AOBJ: 633213).

**Institutional Review Board Statement:** Not applicable.

**Informed Consent Statement:** Not applicable.

**Data Availability Statement:** CML data covering the Netherlands were obtained from https:// data.4tu.nl/authors/Aart_Overeem/10644965, accessed on 15 June 2021, recommended by Aart Overeem. Meterological data of radar and gauge observations were acquired from the KNMI data center archive at: https://dataplatform.knmi.nl, accessed on 15 June 2021. A data-set of CML attenuation covering Israel was graciously offered for research from the Israeli cellphone provider, Pelephone Ltd. Meterological data covering Israel were acquired from the IMS data center, https://ims.data.gov.il/ims/7, accessed on 15 June 2021. The complete R code used in this research is archived in a public repository and will be made available on request.

**Acknowledgments:** The authors wish to express their thanks to Nissim Dvela from Pelephone Ltd. for his assistance and guidance regarding CML data in Israel. Aart Overeem assisted in obtaining the Netherlands CML data, and advised on issues with the RAINLINK package. The authors also thank Yoel Bassin for help in coding. We further express appreciation to the anonymous reviewers whose detailed and appropriate comments led to an improved revision of this paper.

**Conflicts of Interest:** The authors declare no conflict of interest.

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
