# Peer review of "Improved Gridded Precipitation Data Derived from Microwave Link Attenuation"

_remotesensing, doi:10.3390/rs13152953_

Round 1

Reviewer 1 Report

The manuscript deals with the use of microwave link attenuation to improve the precipitation estimate. It describes a methodology for removing links that give erroneous estimates. The methodology is applied to data from Israel and the Netherlands. The text is well structured and the methods and procedures used are explained.

However, the overall idea and logic of the procedure requires a more detailed explanation. In general, there are two basic types of precipitation: convective, which is characterized by large variability in space and time, and stratiform, which is homogeneous in space and does not change significantly over time.

In the introduction, the authors talk about nowcasting and the need to have as much data as possible, which is true. This is especially true for convective precipitation, because in the case of stratiform precipitation, standard data are usually sufficient. However, the authors exclude data from storms that quickly cross microwave links from processing and, in addition, focus on comparing daily totals. In my opinion, this will only reveal systematic errors (probably technical problems) of specific connections, but if these data are used to calculate empirical parameters between attenuation and measured precipitation, then these relationships will be very inaccurate for intense convective precipitation. As a result, I doubt that this additional data will bring improvements for nowcasting and flash-flooding.

The text should make it clear what the aim is. If the goal is to really improve nowcasting, which I don't think so, then the article should focus more on nowcasting.

I also have the following remarks.

The term precipitation grid is not used in meteorological texts. If you want to use this term, you need to say at the beginning what you mean.

I understand the reasons why you use the Kendall correlation coefficient, but it should be noted that it is a coefficient that compares the decrease and increase of values, but does not depend on specific values. The values obtained then do not take into account the actual values. Therefore, it is usual to use standard Pearson correlation, even if the prerequisites for its use are not met. Could you comment this?

Reviewer 2 Report

Improved Precipitation Grids Derived from Microwave Link Attenuation

Micha Silver, Arnon Karnieli and Erick Fredj

General Comments

This research strives to improve interpolated rain grids by correctly identifying problematic links and filtering out those data. Identifying those unreliable links relies on a time-series analysis of CML-derived rainfall compared to weather radar rain grids, using Dynamic Time Warping (DTW).

The paper is extremely well written and easy to read. A very detailed explanation of every step in the methodology is performed, so I have no objections on this subject.

I have a couple of comment I would like the authors could explain better before publication.

1) The methodology used in this paper relies on radar data for a very short period of time, so my concern regarding the flagged (unreliable) links is if this could be applied beyond that time period or if it is just only a ‘snapshop’ for those particular storms. In other words, I am asking about applicability of this technique for other scenario. In case of real-time nowcasting application, do you need to perform this analysis of flagged links every certain period of time of you can use the flagged links during this study? In the first case, this could be troublesome. I guess this issue should be discussed by the authors.

2) In the abstract you claim that for certain storm events the Kendall rank correlation for the set of reliable CML is double that of the complete set, demonstrating that improved precipitation grids can be obtained by removing unreliable links. However, from Table 2 (Israel) and Table 4 (Netherlands), I didn’t see such difference between full and reliable dataset for any storm. What do you mean when you say that Kendall rank correlation for the set of reliable CML is double that of the complete set? Then in the discussion section, you state that small increases in Kendall’s t correlation for the Reliable set, in column 6 of Table 4, should therefore be considered significant, in light of the overall stability of the CML network. From my point of view, it would be nice to test if Kendall rank correlation coefficients are statistically different (and higher for the reliable network) or not for both sites. From the provided information, this is not clear for me.
